# Stopover Ecology of the European Turtle Dove (*Streptopelia turtur*), a Threatened Migratory Bird Species, after the Crossing of an Extended Ecological Barrier

Christos Barboutis [1,2,*], Anastasios Bounas [2], Elisabeth Navarrete [1] and Thord Fransson [3]

1 Antikythira Bird Observatory, Hellenic Ornithological Society/BirdLife Greece, 52 Ag. Konstantinou Str., GR-10437 Athens, Greece
2 Department of Biological Applications and Technology, University of Ioannina, GR-45110 Ioanrnina, Greece
3 Department of Environmental Research and Monitoring, Swedish Museum of Natural History, SE-104 05 Stockholm, Sweden
* Correspondence: cbarboutis@ornithologiki.gr

**Simple Summary:** The European Turtle Dove (*Streptopelia turtur*) is a declining migratory species that overwinters in sub-Saharan Africa and migrates to Europe each spring to breed. Most of the knowledge regarding its wintering grounds, the routes it follows and the sites where it rests and regains mass between migration flights has been obtained from the west and central European populations. In this study, we combined long-term bird ringing data, tracking data and citizen science data to estimate the numbers of Turtle Doves that migrate through Greece every spring and increase our knowledge of how the species uses resting sites after crossing the Mediterranean Sea. Approximately 16% of the European population migrates through Greece every spring, between the end of March and the end of May. On average, Turtle Doves arrive with very low body mass, and some stay on the resting sites for as long as three weeks to regain the mass loss. The extensive use of resting sites after the Mediterranean Sea indicates the probable importance of such sites for the species.

**Abstract:** Migratory routes, important stopover sites and wintering grounds for the Turtle Dove, a declining trans-Saharan migratory bird, are known mainly for populations in western and central Europe, but very little is known about birds using the eastern migration flyway. By combining long-term ringing data, tracking data and citizen science data, a comprehensive picture of the stopover ecology of the Turtle Dove's spring migration in the eastern Mediterranean is presented. Furthermore, a quantitative estimate of the number of birds that migrate over Greece during the spring migration is given. Approximately 16% of the European population migrates through Greece, passing through as early as the end of March, with the passage lasting up to the end of May. On average, the species arrives depleted after the crossing of the Sahara Desert and the Mediterranean Sea, with no systematic refuelling event taking place in North Africa. Both tracking and ringing data indicate that the birds undergo an extensive stopover after the barrier crossing (as much as close to three weeks). Turtle Doves additionally show significant body mass gain during their stay, indicating the potential importance of stopover sites after the Mediterranean Sea for the conservation of the species.

**Keywords:** tracking; migration; islands; Mediterranean

## 1. Introduction

Hundreds of species and billions of individual birds [1–3] cover enormous distances between their breeding and wintering grounds in the African–Eurasian flyway. This flyway is the largest migration system in the world [4], where passerine and near-passerine birds make up the vast majority of the system's migrants [5]. Those biannual seasonal movements involve the crossing of vast ecological barriers, such as the Sahara Desert and

the Mediterranean Sea. In the central [6] and western parts of the eastern flyway (through the Balkan Peninsula) [7], the desert and the sea can be considered one ecological barrier, as there are very few possibilities to refuel during the crossing [8].

The Sahara crossing can take place during a continuous flight [9] or by intermittent flights, with stopovers for mainly resting and recovering [10,11]. When facing the Mediterranean Sea, a bird must cross it in a non-stop continued flight, until the first available land mass [12,13]. Small and large islands scattered in the Mediterranean Sea are often the first available land that migratory birds encounter after crossing the sea and the desert during their northward spring migration.

Migratory birds spend the vast majority of the duration of the migratory journey at stopover sites (e.g., [14–16]). Stopover sites are used heavily for replenishing the fuel load (e.g., [17]), especially during spring after the crossing of the Sahara and the Mediterranean in the central and western parts of the eastern flyway, where a significant portion of birds have been reported to be energy-depleted [18–22]. Nevertheless, the importance of stopover sites, such as islands, just after the barrier crossing is not only limited to refuelling purposes [23–25].

The European Turtle Dove, (*Streptopelia turtur*: hereafter Turtle Dove) is one of the bird species that migrates from its European breeding grounds to the sub-Saharan wintering areas. The species used to be abundant across Europe, Western Asia and Northern Africa [26] but has, in recent years, experienced a rapid decline across Europe [27]. Thus, its extinction risk has increased, being listed as 'Vulnerable' by the IUCN [28]. Detailed data regarding migration routes, stopover locations and wintering sites are available for the western and central European population, while the equivalent information for populations in the eastern part of the breeding range, which constitutes birds using the eastern flyway, is extremely limited [29,30]. During the autumn migration, the species seems to undertake long stopovers in Europe, while during the spring, the species undertakes stopovers in eastern and central North Africa [29,30].

Here, we provide a comprehensive view of the spring migration ecology of the Turtle Dove in the eastern Mediterranean by combining analyses of long-term ringing data, tracking data and citizen science data. We expect that Turtle Doves will show a low arrival body mass, and they will undertake a stopover to recover after the barrier crossing. Furthermore, we provide a quantitative estimate of birds that might use insular or coastal areas of south Greece as stopover sites during the spring migration, and by that, we highlight their conservation value for migratory Turtle Doves during their northward migration.

## 2. Materials and Methods

### 2.1. Study Area

The data for this study were collected on the island of Antikythira (35°51′ N, 23°18′ E), Greece. Antikythira is a small island with an area of ca. 20 km², located between the southern tip of mainland Greece and western Crete (31.5 km southeast of the island of Kythira and at an equal distance northwest of Crete) and 355 km from the North African coast (Figure 1A). Antikythira is part of the Central-Eastern Mediterranean bird migration flyway, and, due to its position [31], it is a is an important migration stopover site for birds during the northward migration in the spring [21], where bird species seem to stopover from just a few to several days (e.g., [22,32]).

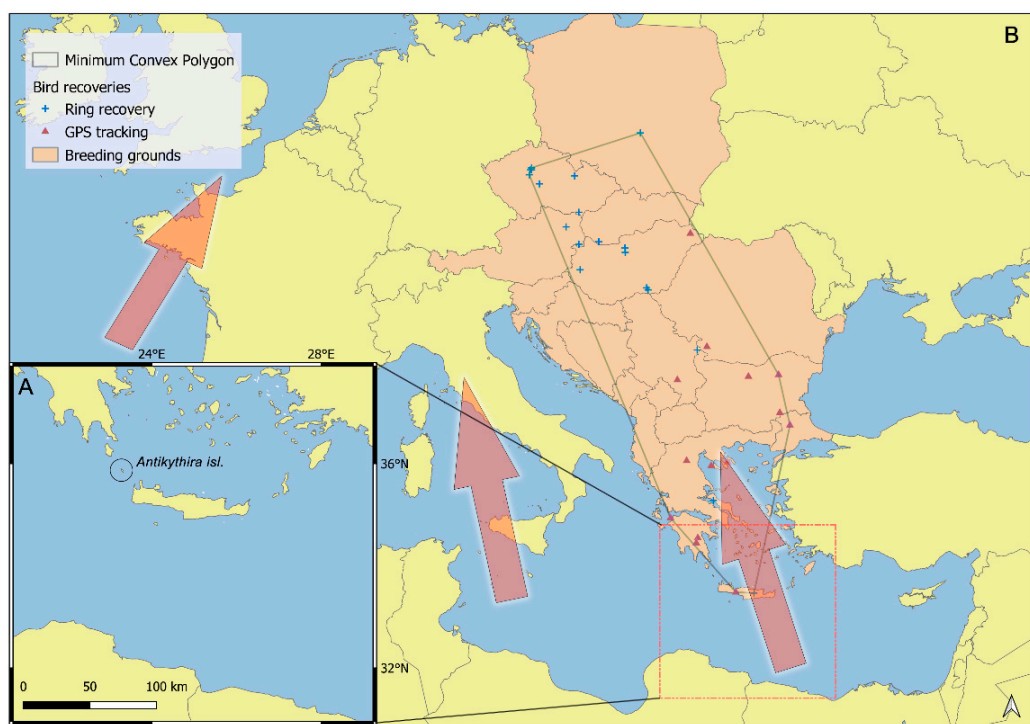

**Figure 1.** (**A**) Location of our study site in relation to continental Europe and the North African coast. (**B**) Convex polygon around the recoveries and the putative breeding grounds of the Turtle Doves that migrate through Greece during spring. Arrows show the three flyways used by the species, as described by Spina et al., 2022 [33].

### 2.2. Bird Captures and Handling

Turtle Doves at Antikythira were captured within the Antikythira Bird Observatory, a monitoring project of migratory birds, run by the Hellenic Ornithological Society and the Hellenic Bird Ringing Centre, which takes place between the end of March and the end of May. Mist netting (16 × 16 mm mesh, nylon) at Antikythira took place every day from dawn and thereafter for eight hours, except for days with adverse weather conditions. The fixed total length of the mist nets used every season was 126 m. Trapped birds were aged and sexed according to Baker [34] and Demongin [35] and weighed to the nearest 0.1 g. The maximum wing length [36] was recorded as a measurement of size. The trapping effort barely differed between the years regarding both the total length of the nets and the positions of mist nets used. We assumed that the birds were first captured soon after arrival on the island, and the body mass recorded at ringing was used as the arrival body mass.

### 2.3. Statistical Methods

The phenologies of the species' spring migration through Greece were estimated using all the available data from eBird (www.ebird.org (accessed on 6 December 2022); Sullivan et al. [37]). We used the R package auk [38] for extracting and processing the eBird data. Specifically, we kept only the spring observations (March to May) from the south Aegean and Ionian Seas (spatial bounding box 20.9, 34, 28.5, 37.8) to retrieve records of putatively migrating birds only. We restricted the observations to the standard travelling and stationary count protocols and only retained complete checklists. The observation dates were transformed to days from the first of January in order to estimate the first passage and median dates (1st, 3rd quartiles) of the species through Greece. The first capture and median dates (1st, 3rd quartiles) were additionally estimated based on the ringing data (from 2007 to 2022). Likewise, the capture dates were transformed to days from the first of January and the median dates were measured as the dates when the cumulative number of ringed birds passed 50%.

The stopover duration was estimated using both GPS (16) and radio transmitters (7) deployed between 18 April 2017 and 15 May 2022 in Antikythira using licences issued by the Hellenic Ministry of Environment and Energy. In more detail, 11 PinPoint GPS Argos Solar tags (Lotek Wireless Inc., Newmarket, ON, Canada, tag mass of 5.5 g), 5 HQBG0804 GPS/GSM tags (Hunan Global Messenger Technology Co. Ltd., Xiangtan, China, tag mass of 5.5 g), 2 Pip Ag376 radio tags (Biotrack Ltd., Wareham, UK, tag mass of 1.5 g) and 5 LifeTag radio tags (Cellular Tracking Technologies Inc., Rio Grande, NJ, USA, tag mass of 0.45 g) were used. The minimum stopover duration—the average number of days between the first and last capture of the recaptured birds—was additionally used in order to compare the results of the two methods. Furthermore, proxy for the migration direction of the species were estimated using the headings between Antikythira and the landing location after the first night of flight (data from the GPS/GSM tags). Uniformity of directions was tested with the Rayleigh test of uniformity.

We used the data from tracked birds to explore the relationship between a bird's decision to remain or depart from the stopover site and its arrival body mass (assuming that the captured birds newly arrived) by means of a logistic regression in R [39]. The R package cutpointr [40] was then used to perform an optimal cut point analysis using the Youden index as a metric to identify the birds likely to stop over on the island, as predicted by their initial body mass value. The mean body mass change (mass gain/minimum stopover duration) was estimated using the data from birds that were retrapped at our study site after at least one day.

In order to make a rough quantitative estimate of the number of Turtle Doves migrating through Greece every spring, we used the recent breeding bird population estimates of the species [41] following the approach of Hahn et al. [2]. Specifically, we used our GPS tracking data along with ring recoveries (provided by the Hellenic Bird Ringing Centre) and drew a minimum convex polygon around the observations that encompassed the breeding grounds of the Turtle Doves migrating through Antikythira. Then, we used the breeding population estimates for each country that was included in the polygon as an estimate of the number of birds that would migrate through Greece. As it is suggested that birds following the central flyway adopt a clockwise loop migration [30], only the recovery data of the Turtle Doves that were ringed in Greece during the spring and recovered during the breeding season and data from birds that were ringed during the breeding seasons and recovered during spring in Greece were used.

## 3. Results

In total, 670 Turtle Doves were trapped during the spring season between 2007 and 2022. The median date of the spring passage based on the ringing data was 28 April (1st, 3rd quartiles: 21 April, 5 May), with the earliest bird trapped on 3 April 2013 and the latest on 24 May 2019. Likewise, a total of 114 eBird records from 2011 to 2022, comprising 357 Turtle Doves, were used. The median date of the spring passage based on the observation data was 25 April (1st, 3rd quartiles: 21 April, 29 May), with the earliest bird observed on 29 Mar 2016 and the latest on 29 May 2022.

The mean arrival body mass of the Turtle Doves at our study site was 120.6 ± 15.94 g (n = 628), ranging from 82.9 to 190.7 g. A total of 20 out of 670 Turtle Doves (3%) ringed were recaptured after at least one day after the initial trapping event. The minimum stopover duration for the retrapped birds was 6 ± 4.8 days (n = 20), varying from 2 to 21 days. Using the tracking data, the mean stopover duration of tracked birds was 4 ± 6.2 days (n = 23), varying from 0 to 17 days. Of the birds that had a stopover of 1 day or more at our study site, the mean stopover duration was 8 ± 6.9 days (n = 12), varying from 1 to 17 days. During their stopover, the Turtle Doves gained on average 8.8 ± 11.7 g (n = 16, 3.0 ± 4.9 g/day), varying from losing as much as 11.1 g to gaining up to 34 g. The logistic regression model (Table 1) showed that the decision of the tracked birds to remain in the stopover decreased by 7% for each additional gram of body mass gained (*p* = 0.006). The optimal body mass cut point differentiating each decision (remain vs. depart) was found to

be 140.6 g, achieving a Youden index maximum value of 0.54 (sensitivity 63.6%, specificity 55.8%; Figure 2). We calculated that 88.8% of the Turtle Doves (595 out of 670 trapped birds) arriving at our study site have a body mass of less than 140.6 g (Figure 3).

**Table 1.** Results of the binary logistic regression analysis between a bird's decision to remain or depart from the stopover site and its arrival body mass.

| Predictor | β | SE β | z Value | df | p | e$^\beta$ (Odds Ratio) |
|---|---|---|---|---|---|---|
| Constant | 9.48 | 3.18 | 2.98 | 40 | 0.003 | NA |
| Mass | −0.07 | 0.02 | −2.75 | 40 | 0.006 | 0.93 |

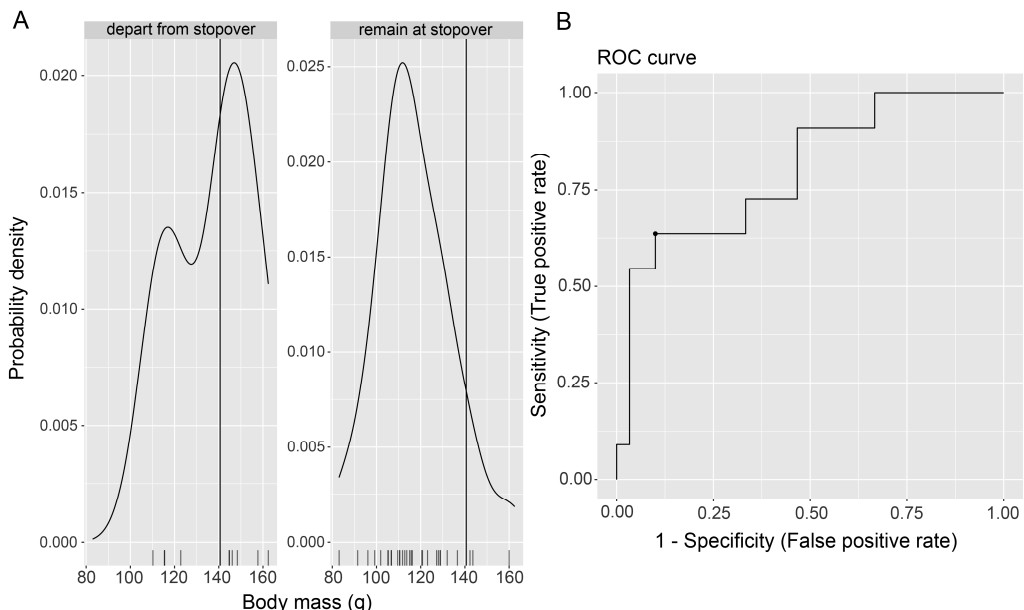

**Figure 2.** Output of cutpointr, used to identify birds likely to stopover on the island, as predicted by their initial body mass value. (**A**) Probability density function (probability per body mass unit) of birds that depart or remain at the stopover. Vertical black lines indicate the optimal body mass cut point value of 140.6 g. (**B**) ROC curve for the selection of the best body mass cut-off.

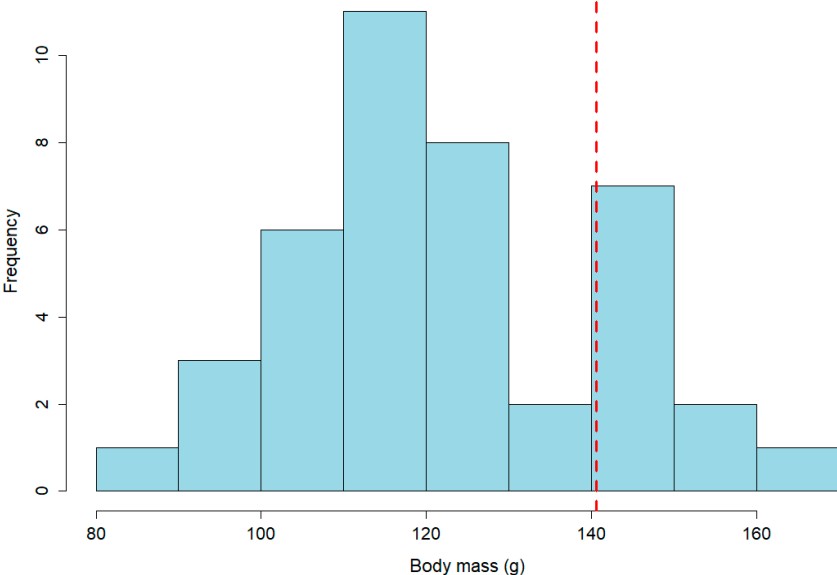

**Figure 3.** Distribution of arrival body mass (body mass at first capture) in relation to the threshold departure body mass of 140.6 g (dashed line).

Birds taking off from Antikythira are heading on average north ($\alpha = 359°$, r = 0.84, n = 16, *p* < 0.001, Rayleigh test), ending up at breeding grounds located in at least 15 countries, based on ring recoveries and GPS/GSM tags (Figure 1B). Based on our conservative approach, the number of Turtle Doves that migrate through Greece during the spring is on average 1,543,300 (ranging from 950,600 to 2,136,000). This number of birds represents 14.9% to 18.1% of the European population and 2.2% to 3.7% of the global breeding population, respectively.

## 4. Discussion

Turtle Doves arrive in Greece as early as the end of March, with the passage lasting up to the end of May. The median date of their passage is taking place during late April. The first birds pass through the study area in early April and the numbers then increase steadily, peaking during the end of April/early May and decreasing rapidly from mid-May onwards. Interestingly, both the systematic long-term bird ringing data and direct observation citizen science data reveal the same results, thus confirming the potential of such opportunistic data to effectively address questions regarding bird migration [37]. The observed spring migration passage of the species in Greece is similar to that reported for adjacent regions and more western regions of the Mediterranean [26,42–44].

There is evidence that stopover sites and habitats are selected or avoided during migration [45–48]. The relatively large number of depleted birds observed in the eastern flyway after the sea crossing during the spring migration [21] indicates that at least a considerable portion of these birds choose their stopover sites, not based on a selection process but rather as a necessity. Birds arrive at our study site with an average body mass of 120 g, which is within the lower range of values reported for the species in localities just after the crossing of the Mediterranean Sea [26,44,49]. The arrival body mass appears to be larger in the west compared to the central and eastern Mediterranean. This could be either due to the larger stretches of the sea the birds need to cross or the less extensive use of stopovers and refuelling in North Africa [29,30] over the eastern flyway. Unfortunately, a lack of tracking data on birds using the eastern flyway during the spring does not allow for any evaluation of the extent of the stopover sites that are used in North Africa. Nevertheless, the mean arrival body mass indicated that, on average, the birds arriving at our study site have not refuelled extensively before the sea crossing.

Once arriving at our study site, both the ringing data and tracking data indicate that the Turtle Doves follow a mixture of strategies, with some birds using the site as a resting point and resuming their migratory journey the following night or after one day, and others undertaking an extensive stopover. The fact that some of the birds with transmitters attached resume migration the following night, only using the site as a resting point, supports the assumption that the birds are first captured soon after arrival. Although some individuals might refuel before the sea crossing, the stopover sites serve multiple functions, for example, for physiological recovery, the avoidance of adverse environmental conditions and predation risks and refuelling (reviewed in [50]), and this could at least partially explain the large variation in the stopover duration. Surprisingly, despite the cost involved in a prolonged stopover duration [51–53], some of the birds that do stopover make an extensive stop that could reach up to three weeks, based on both the ringing and tracking data. Based on the logistic regression on the stopover duration and arrival body mass, most of the Turtle Doves arriving at our study sites will stay on the island for at least one day. This is in contrast to the low recapture rate (3%), but the monitoring methods applied (mesh size of mist nets) are not ideal for the species and may have affected the estimates based on ringing effort.

Although migratory birds with narrow or very specific habitat requirements during their breeding or wintering grounds have been shown to be able to successfully use a much wider spectrum of habitats during a stopover [54], the available habitats at our study site [55] approach the needs of the species during the breeding period [56,57]. This is also reflected in the relatively high rate of body mass gain the Turtle Doves experience

at the site compared with the equivalent values from the western Mediterranean [44]. The elevated refuelling rate, extended stopover duration, average arrival body mass and threshold departure body mass indicated by the logistic regression suggests that at least a portion of the Turtle Doves use the study site as a refuelling area, and, thus, no extensive refuelling takes place in eastern North Africa, as seems to happen in the west after the desert crossing [29,30]. Nevertheless, refuelling in eastern North Africa is possibly much more challenging as there are much less suitable habitats available compared to west North Africa [58], and crossing the Sahara desert and the Mediterranean Sea without refuelling might be more a necessity rather than a strategy for the Turtle Doves in this area. It has been shown that Turtle Doves breeding in France use stopover sites in Northwest Africa for several weeks in the spring before they continue to their breeding sites [29]. A similar pattern, with a prolonged stopover in the spring in Northwest Africa, has also been found in Northern Wheatears (*Oenanthe oenanthe*), and it has been suggested that this might be a strategy of using areas with optimal foraging conditions at a time when foraging conditions in the wintering areas just south of the Sahara desert have deteriorated [59]. The Sahel region, south of the Sahara desert, receives some rain from July to September and dries out during the rest of the year [43]. It might very well be a part of a strategy for some Turtle Doves following the eastern flyway to leave the wintering areas before the conditions have become too harsh and make a prolonged stopover in Greece, at about the same latitudes as in Northwest Africa, before continuing to breeding sites further north in Eastern Europe.

The estimation of the number of Turtle Doves migrating over Greece during the spring, even though rough, demonstrates the magnitude of the phenomenon. Given the small extent of bird ringing and, consequently, recoveries from Greece [60], and the few available tracking data, the estimation might even be conservative, as migratory Turtle doves migrating through Greece might be heading to breeding grounds beyond the ones estimated in this study. Nevertheless, to the extent of our knowledge, this is the first attempt to quantify the magnitude of the spring migration of Turtle Doves over Greece. Even if the estimate is rough, it seems that a significant number of migrating Turtle Doves, comprising a substantial percentage of the European population, will end up in Greek islands and coastal areas every spring after crossing the Mediterranean Sea. As indicated from our results, a portion of those birds arrive energy-depleted and will stop over for several days before resuming their migration journey to their breeding grounds. Turtle Doves in need of swift energy replenishment are pressured by rapid urbanisation, agricultural intensification and widespread tourist activities at stopover sites in insular and coastal areas in Greece and the Mediterranean as a whole [61]. On top of the ongoing environmental degradation, birds are facing significant spring poaching pressure in the Ionian Islands and on the southwest coast of Greece [62,63]. Increased drought events and pressure from changes in land use taking place throughout the Mediterranean basin [64,65] will probably result in additional pressure on migratory birds. Drier island and coastal sites in Greece and the whole Mediterranean would probably result in slower refuelling rates, prolonged stopover durations after a sea crossing and/or the introduction of new stopover sites further north, possibly facilitating a mismatch between the time of arrival and breeding [66,67].

Regarding conservation actions and strategies that have, until now, almost exclusively focused on breeding grounds [68,69], focusing on important stopover sites for Turtle Doves would benefit their overall fitness. Detailed tracking data on stopover use and the connectivity between breeding sites, stopovers and wintering grounds, as well as monitoring schemes to examine how migratory birds manage to cross the Sahara Desert and the Mediterranean Sea, are therefore of utmost importance in order to inform conservation strategies [70,71].

**Author Contributions:** Conceptualization, C.B. and A.B.; Methodology, A.B. and C.B.; Software, A.B.; Validation, C.B., A.B. and E.N.; Formal Analysis, C.B.; Investigation, C.B. and A.B.; Resources, C.B.; Data Curation, C.B. and A.B.; Writing—Original Draft Preparation, C.B. and A.B.; Writing—Review and Editing, T.F. and E.N.; Visualization, A.B.; Supervision, T.F.; Project Administration, C.B.; Funding Acquisition, T.F., A.B. and C.B. All authors have read and agreed to the published version of the manuscript.

**Funding:** This work is part of ROUTES research project, supported by the Hellenic Foundation for Research and Innovation (H.F.R.I.) under the "2nd Call for H.F.R.I. Research Projects to support Post-Doctoral Researchers" (Project No. 805). We are very grateful for the financial support from the Marie-Claire Cronstedts Foundation and the Swedish Museum of Natural History for the satellite transmitters used. Antikythira Bird Observatory is funded by the A.G. Leventis Foundation.

**Institutional Review Board Statement:** Bird ringing and tagging was done under the licences 6ΥΟΞ4653Π8-ΞΓ5, ΨΑΤΧ4653Π8-Η6Τ, 6ΥΙΒ4653Π8-ΣΑΛ, ΥΠΕΝ/ΔΔΔ/6117/170 issued form the Hellenic Ministry of Environment and Energy.

**Data Availability Statement:** The data may be available from the authors upon request.

**Acknowledgments:** The fieldwork was carried out with the relevant permits from the Hellenic Ministry of Environment and Energy. Rings were supplied free of charge by the Hellenic Bird Ringing Centre. We thank the numerous volunteers of Antikythera Bird Observatory. This is contribution No. 37 from the Antikythira Bird Observatory, the Hellenic Ornithological Society/BirdLife Greece.

**Conflicts of Interest:** The authors declare no conflict of interest. Publishing the results was a precondition for the funding. The funders had no role in the design of the study; in the collection, analyses or interpretation of data; in the writing of the manuscript; or in the decision to publish the results.

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
