# Peer review of "Stopover Ecology of the European Turtle Dove (Streptopelia turtur), a Threatened Migratory Bird Species, after the Crossing of an Extended Ecological Barrier"

_2673-6004, doi:10.3390/birds4020017_

Round 1

Reviewer 1 Report

A well written (some grammatical errors and miss-spellings) manuscript, but it still needs some clarification and additional detail especially in the methods section. Please also add a discussion on the assumption that first caught birds recently arrived. The logistic regression analysis is not clear, please clarify and adjust results/conclusions where needed. I provided some additional comments in the attached document. 

Author Response

Dear Reviewer

On behalf of all the authors i hearby, provide the reply to your comments

A well written (some grammatical errors and miss-spellings) manuscript, but it still needs some clarification and additional detail especially in the methods section. Please also add a discussion on the assumption that first caught birds recently arrived. The logistic regression analysis is not clear, please clarify and adjust results/conclusions where needed. I provided some additional comments in the attached document.

-a figure showing the different flyways would be good

We added the flyways in the map.

-P2 “ The trapping effort barely differed between years” : vague, please provide more information

Addition in the method section clarifies the sentence.

-using all available data from e-bird - which years?

We inserted the years in the results section

-Stopover duration was estimated by using both GPS (16) and radio transmitters- please add some info about age and sex. Did you see differences between ages and sexes? why did you not include such analyses?

We did not find significant differences between age groups. We decided not to include any comparison between age and sex as the sample size is not enough to give valid conclusions.

- We explored the relationship between a bird's decision to stopover or depart from our study site the next day after capture and its arrival body mass - how exactly did you do this as you ahve no data for birds that did not stopover, so i am uncertain you can do this analyses in the right way.

The Reviewer is right to be confused, we poorly phrased the method description. What we meant is that we used a logistic regression to explore whether a bird will remain or depart from the stopover site (based on data from birds that arrive at the stopover site). We know this decision as the analysis included all tracked birds. We rephrased.

-what did you do when birds were retrapped multiple times?

Sorry again, as explained above the regression is based on decisions of tracked birds not of ringing data. Therefore we know which birds remained or departed and were able to associate this with arrival body mass.

-elsewhere?

Indeed, the word is not needed

Results

-how many did not stop or stopped less than a day?

Out of 23, 11 stopped for less than a day.

-confusing as you write more than one day in the previous sentence

This has been revised to “birds that stopover for a day or more”.

-confusing as you can only measure their weight when they do stopover. please clarify

Hopefully, its clearer now that we addressed this misunderstanding in the methods

-We calculated that 88.8% of Turtle Doves arriving at our study site have a body mass less than 140.6 g - how did you calculate this?

By means of the logistic regression we calculated that the cutpoint for remaining or departing from the stopover was 140.6 g . Then we went back on the data (670 Turtle Doves ringed, as we already describe) and checked how many of them have a mass under the remaining threshold. 88.8% of those did. We added a small statement and we believe that it is more understandable after the explanation of the logistic regression method.

-Fig2 - adjust axes titles to reflect what actually is shown (e.g. weight in grams)

We revised the Figure according to EIC suggestions.

-Arrival body mass assumes that birds are first captured soon after arrival. Discuss this and the implications in the discussion

We have now included in the method section that we assume that birds are first trapped soon after arrival. The fact that about half of the birds attached with transmitters resume migration the following night gives support for this assumption and a sentence about this has also been included in the discussion.

Discussion

- Arrival body mass appears to be larger in the west compared to the central and eastern Mediterranean - what is the difference?

As we have written, it is in the lower range of what has been reported from other sites and we give three references to this. 

- Based on the logistic regression on stopover duration and arriving body mass, most of the Turtle Doves arriving to our study sites will stay at the island for at least one day - So better explain the analyses, as I am not sure this has been correctly done.

We believe that the analysis is much clearer after the changes we made.

Sincerly

Dr. Christos Barboutis

Antikythira Bird Observatory,

Hellenic Ornithological Society/BirdLife Greece

Reviewer 2 Report

Hello! This paper combined individual tracking, citizen data, and bird ringing data to comprehensively analyze the importance of the island of Antikythira to the threatened migratory bird, the European Turtle Dove, and it was proved that this area was an important staging sites for doves after crossing the ecological barrier. The methods are various and the data are credible. As an important basic data for scientific research, I think it can meet the requirement of publication.

One of the key conclusions of this paper, “Approximately 16% of the European population migrates through Greece every spring”, does not provide sufficient data analysis and argumentation methods. This is where the author needs to be more specific about how he arrived at this conclusion.

Author Response

Dear Reviewer

On behalf of all the authors i hearby, provide the reply to your comments

Hello! This paper combined individual tracking, citizen data, and bird ringing data to comprehensively analyze the importance of the island of Antikythira to the threatened migratory bird, the European Turtle Dove, and it was proved that this area was an important staging sites for doves after crossing the ecological barrier. The methods are various and the data are credible. As an important basic data for scientific research, I think it can meet the requirement of publication.

One of the key conclusions of this paper, “Approximately 16% of the European population migrates through Greece every spring”, does not provide sufficient data analysis and argumentation methods. This is where the author needs to be more specific about how he arrived at this conclusion.

We have provided some more details in the methods to facilitate interpretation of this estimate.

Thank you again for the opportunity to publish in your Journal

Sincerly

Dr. Christos Barboutis

Antikythira Bird Observatory,

Hellenic Ornithological Society/BirdLife Greece